# Recent Advances in Anomaly Detection Methods Applied to Aviation

**Luis Basora** *,†,‡, **Xavier Olive** †,‡ **and Thomas Dubot** †

ONERA DTIS, Université de Toulouse, CEDEX 4, 31055 Toulouse, France; xavier.olive@onera.fr (X.O.); thomas.dubot@onera.fr (T.D.)

* Correspondence: luis.basora@onera.fr
† Current address: 2 avenue Édouard Belin, CEDEX 4, 31055 Toulouse, France.
‡ These authors contributed equally to this work.

**Abstract:** Anomaly detection is an active area of research with numerous methods and applications. This survey reviews the state-of-the-art of data-driven anomaly detection techniques and their application to the aviation domain. After a brief introduction to the main traditional data-driven methods for anomaly detection, we review the recent advances in the area of neural networks, deep learning and temporal-logic based learning. In particular, we cover unsupervised techniques applicable to time series data because of their relevance to the aviation domain, where the lack of labeled data is the most usual case, and the nature of flight trajectories and sensor data is sequential, or temporal. The advantages and disadvantages of each method are presented in terms of computational efficiency and detection efficacy. The second part of the survey explores the application of anomaly detection techniques to aviation and their contributions to the improvement of the safety and performance of flight operations and aviation systems. As far as we know, some of the presented methods have not yet found an application in the aviation domain. We review applications ranging from the identification of significant operational events in air traffic operations to the prediction of potential aviation system failures for predictive maintenance.

**Keywords:** anomaly detection; aviation; trajectory; time series; machine learning; deep learning; predictive maintenance; prognostics and health management; condition monitoring; air traffic management

---

## 1. Introduction

### 1.1. Anomaly Detection

Anomaly detection is an active area of research encompassing a significant number of techniques developed in diverse fields such as statistics, process control, signal processing and machine learning. The goal is to be able to identify data deviating from or not being in agreement with what is considered normal, expected or likely in terms of the data probability distribution, or the shape and amplitude of a signal in time series.

Another commonly used term for anomaly is *outlier* and both are sometimes used interchangeably. Also, Pimentel [1] prefers the term *novelty detection* to anomaly detection when the goal is to identify data differing in some degree from the data previously observed, even though the underlying detection methods are often the same. The distinction between novel data and anomalies is that the former is usually considered as normal data after being detected [2].

One of the main challenges in anomaly detection is the difficulty to clearly distinguish normal instances from anomalous ones, as the boundary between the two is usually imprecise and evolves

over the time in some application domains. In addition, anomalies are often rare events, so labelled datasets for model training and validation are either unavailable or severely imbalanced in favor of normal instances. As a consequence, semi-supervised or unsupervised learning is more frequently used than supervised learning.

In a semi-supervised approach for anomaly detection, it is assumed that in the training set labels exist only for the normal class. If there are any unlabelled instances, then they are assumed to have the same label as the labelled instances that are nearby in terms of distribution. In any case, as the training data should contain a vast majority of normal data, the model is actually trained to learn the normal behaviour of a system and then used in the test phase to identify anomalies.

On the other hand, with unsupervised learning techniques, the only assumption is that a very small fraction of the overall data is anomalous data or that anomalies lie far away from most data instances or in low-density regions. A final consideration is that even though anomaly detection is often based on unsupervised learning, Erhan et al. [3] explain how unsupervised methods can be of significant help in building supervised predictive models.

Chandola et al. [2] identify three main types of anomalies:

- *Point anomalies.* A data point that differs significantly from the rest of the data points in the dataset considered. For instance, in a time series of French temperatures in summer, a temperature of 40 °C can be considered as an anomaly even with the undergoing climate change.
- *Contextual anomalies.* When a data point is an anomaly only in a particular context. The context is defined by the contextual attributes, which usually refer to time (time series) or location. For instance, in a time series of summer temperatures by country, a temperature of 40 °C is an anomaly in France, but it might be not in hotter countries like Libya where temperatures in summer are commonly around 40 °C. Attributes (e.g., temperature) indexed by contextual attributes (e.g., country) are called behavioural attributes. Not only anomalies in spatial data but also in time series fall into this category, e.g., 40 °C can be an anomaly in Libya from October to April, as at this time average temperatures range from 15 °C to 30 °C.
- *Collective anomalies.* When a group of data in a dataset is an anomaly as a whole, but the individual instances in that group (or subsets of them) might be not on their own. In time series, this would correspond for instance to a situation or condition persisting over an abnormal long time. Collective anomalies can only be detected in datasets where data is related someway, i.e., sequential, spatial or graph data.

Detection techniques for contextual and collective anomalies are particularly relevant in our survey since they are applicable to time series data. The adoption of a particular method depends on the nature of the anomaly, the characteristics of data (existence of labels, number and types of data attributes, data volume) and the expected output (label or score, need for result interpretability). For instance, the lack of labels or the presence of just normal data in the training set requires unsupervised or semi-supervised learning techniques. On the other hand, different statistical models or distance functions are used for continuous or categorical data. As another example, some techniques do not work well with high-dimensional data, e.g., data sparsity can be a real issue for both statistical and clustering techniques: the amount of data needed for statistical significance grows exponentially with the dimensions and data instances appear all far away from each other. Finally, the adoption of a particular method will also depend on whether the domain experts require an understanding of how a model produces the results. If so, methods learning human-readable logical expressions from data such as temporal logic-based models are a better option than black-box models such as neural networks.

*1.2. Previous Surveys on Anomaly Detection*

Previous surveys in the literature offer a comprehensive and structured review of anomaly detection methods. A first survey in two parts has been published in 2003 by Markou and Singh [4,5],

the first focusing on the statistical approaches and the second one on neural networks approaches. Chandola et al. [2] provided in 2009 a good understanding of the subject and a relevant taxonomy of the different techniques. A more recent and extensive survey on novelty detection by Pimentel et al. [1] provides more than 300 references classified in five main categories.

More specific surveys, as Zimel et al. [6] (2012), focus on the challenges of unsupervised outlier detection algorithms applied to high-dimensional data. Aggarwal [7] (2013) reviews the techniques in the literature for outlier ensembles and the principles underlying them. Xu et al. [8] (2019) provide a more recent review on the progress made in anomaly detection with a focus to high-dimensional and mixed types. On a side topic, Längkvist et al. [9] (2014) present a more general review on unsupervised machine learning applied to time series. Akoglu et al. [10] provide a general overview of the state-of-the-art methods for anomaly detection in graph data.

### 1.3. Motivation and Organisation of the Survey

The complexity of aviation systems and traffic operations makes the use of model-based anomaly detection techniques difficult due to insufficient model fidelity and over-simplified assumptions. Indeed, a significant research effort have been dedicated to the development of data-driven approaches, which have notably benefited from significant advances in machine learning and the availability of massive amounts of sensor-generated data. However, some of the classical statistical and machine learning techniques for anomaly detection do not scale well with large datasets or perform poorly with high-dimensional data, which is usually the kind of data available in aviation. In this context, recent advances in the deep learning field should significantly improve the performance of anomaly detection with large-scale high-dimensional data.

The motivation of this survey is to review the state-of-the-art in data-driven anomaly detection methods and their application to the aviation domain: special attention is given to the techniques applicable to large-scale high-dimensional time-series data, i.e., flight trajectories and sensor-generated data for prognostics and health management (PHM) purposes, widely applied in the predictive and condition-based aircraft fleet maintenance.

Recent advances in neural networks and deep learning as well as on anomaly detection using temporal logic based learning justify an up-to-date review of the taxonomy of classical anomaly detection techniques covered in the previous mentioned surveys. This need has been recently addressed in part by Chalapathy et al. [11] with a detailed survey on the state-of-the-art of deep-learning based anomaly detection, but only for domains other than aviation. Concerning the aviation domain, the survey of Gavrilovski et al. [12] focuses indeed on data-mining anomaly detection techniques specifically applied to flight data, but does not cover any of the recent advances on anomaly detection.

Therefore, the goal of the present survey is to complete the previous contributions by proposing a review of anomaly detection techniques applied to aviation, including the recent advances on neural networks and deep learning as well as temporal logic based learning. The review of the recent advances on temporal-logic based learning offer a more complete picture of the available anomaly detection techniques by providing an alternative to black-box models for applications where domain experts need to be able to interpret the results.

This contribution is organised as follows. Section 2 reviews the big picture of the already published surveys and the taxonomies used for grouping the main anomaly detection methods. Section 3 reviews the latest publications with a particular focus on categories of methods that recently become popular, namely recurrent neural networks (Section 3.1), convolutional neural networks (Section 3.2), autoencoders (Section 3.3), generative models (Section 3.4) and temporal-logic based learning (Section 3.5). Section 4 focuses on how these data-driven methods have been recently employed on two aviation-related domains of application, namely the identification of significant flight operational events in air traffic operations (Section 4.1) and the prediction of aviation system faults for predictive maintenance (Section 4.2).

## 2. Taxonomy of Classical Methods in Previous Surveys

In this section, we introduce some of the classical anomaly detection techniques already reviewed in previous surveys (see Figure 1). We focus on the main methods, in particular the ones that have been applied to aviation. For a more extensive review, the reader is referred to the previous surveys.

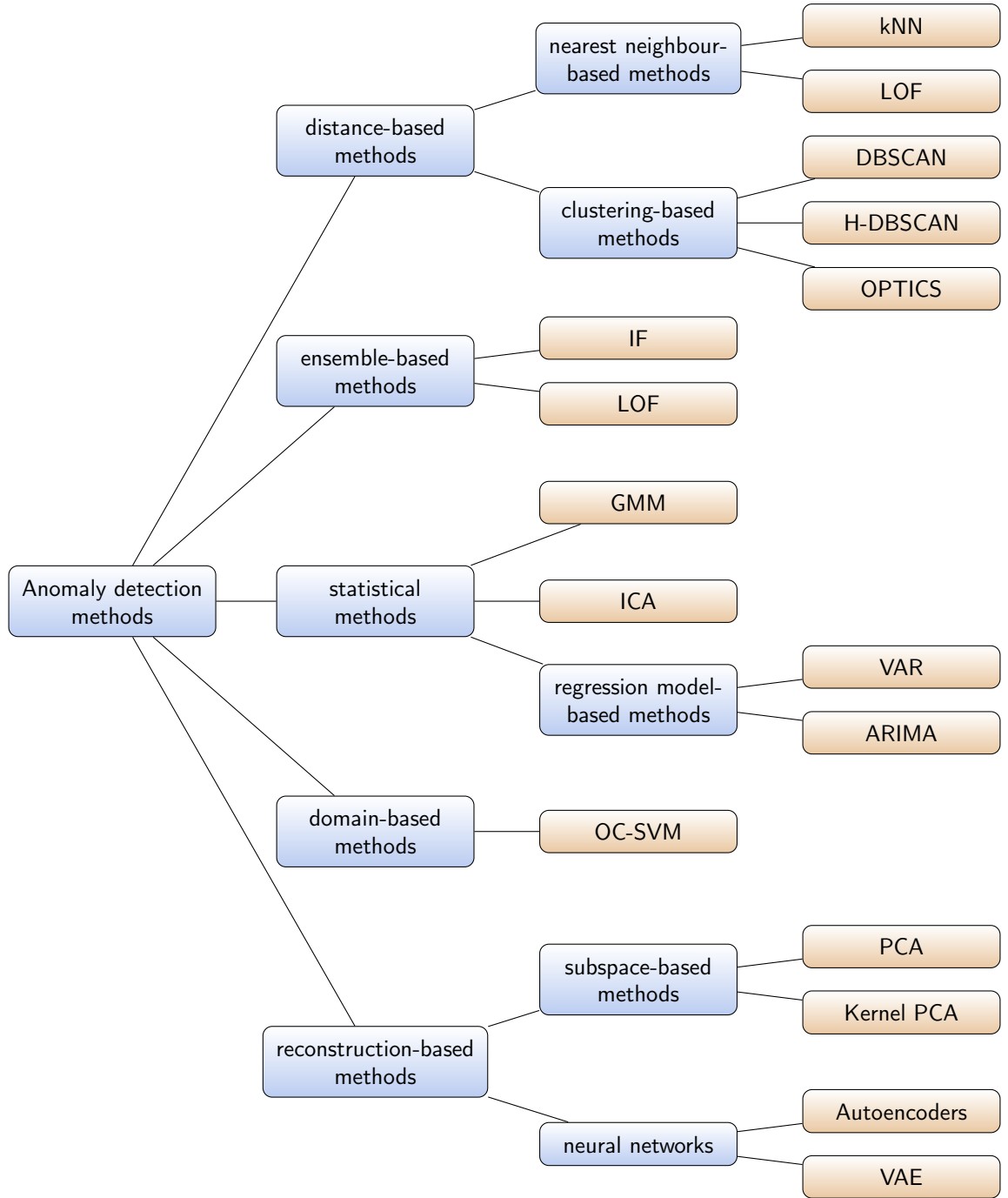

**Figure 1.** Taxonomy of classical anomaly detection methods.

### 2.1. Distance-Based Methods

This category identified by Pimentel et al. [1] includes both nearest neighbour-based and clustering-based anomaly detection approaches, which are two approaches identified as two separated main categories in the taxonomy by Chandola et al. [2]. All the methods in this group rely on the definition of a distance/similarity function between two data instances, which is not always evident

when data instances are not points but more complex data like time series. Most of the techniques discussed here do no require the distance/similarity function to be strictly a metric, but at least to be positive-defined and symmetric (triangle inequality not required).

2.1.1. Nearest Neighbour-Based Methods

In this category, we include the methods capable of detecting an anomalous data point based on either its distance to the neighbour points or its relative data density.

One of the basic distance-based techniques is the k-Nearest Neighbours (kNN) method in which an anomaly score is computed for each data instance defined as the distance to its k-Nearest Neighbours. Then, a threshold is used to determined whether a data point is anomalous or not. Several variants of this technique exist to deal with different data types (continuous and discrete) by using different distance/similarity functions, to compute the scores differently or to improve the complexity of the basic algorithm which is $O(n^2)$ (where $n$ is the data size).

Density-based approaches assume that density around an outlier or anomalous point should be significantly lower than the density around a normal data point. For instance, the Local Outlier Factor (LOF) [13] method computes the densities of the (k-nearest) neighbours and the anomaly score of an instance is the ratio between the local density of the instance and the average of the densities of its neighbours. LOF performs relatively well to detect very sparse anomalies among a large volume of normal data such in the case of network intrusion attacks [14]. This makes it also potentially practical for some of the aviation applications.

In fact, one improved variant of LOF, called Local Outlier Probability (LoOP) algorithm [15], is used by Oehling et al. [16] to look for rare safety events in large amounts of sensor-generated flight data. The main advantage of LoOP is that it provides a score which can be directly understood as the probability for a data instance to be an outlier. This standardized outlier score allows for comparisons over one dataset and even over different datasets. The probabilistic approach of LoOP is also more reliable and tolerant to noise than LOF, where an inappropriate choice of parameter $k$ can cause unstable results.

Although LOF works better than kNN with datasets of varying density, both techniques do not scale well with large and high-dimensional data as they need to compute pairwise distances between data points and determine the nearest-neighbours. In fact, the computational complexity not only can be high during the training but also during the testing phase. See [1,2] for a more exhaustive survey on the multiple improved variants reducing the quadratic complexity of the basic techniques. For instance, the Orca program [17] is an example of an improved variant often cited in aviation papers as part of the literature review in anomaly detection.

2.1.2. Clustering-Based Methods

Clustering is a well-known unsupervised and semi-supervised technique to group similar data instances into clusters based on the definition of a pairwise distance or similarity function. Clustering-based anomaly detection will be introduced in slightly more detail because it is widely used to identify relevant flight operational events [18–22] (see Section 4.1.2).

Chandola et al. [2] distinguish three different categories of clustering-based anomaly detection techniques:

- In the first category, the techniques assume that normal data instances belong to a cluster whereas anomalies do not: anomalies correspond instead to the so called clustering *outliers* or noise. Thus, any clustering algorithm that does not force all data instances to belong to a cluster can be used. The most popular ones are density-based clustering algorithms such as DBSCAN [23], HDBSCAN [24] or OPTICS [25].
- In the second category, the assumption is that normal instances are near their closest cluster centroid whereas anomalies lie far away from them. In this case, two steps are required for

anomaly detection: run an algorithm to cluster the data, and then compute an anomaly score for each data instance based on the distance to its closest cluster centroid. An example of technique in this category often cited in aerospace papers is the Inductive Monitoring System (IMS) algorithm [26].

- The third category addresses the issue with the methods in the two previous categories when clusters of anomalies are formed. This is because the assumption is now that normal instances belong to large and dense clusters and anomalies to sparse or small clusters. A threshold is thus defined on the cluster size or density to determine the anomaly cases.

Some clustering methods are designed to support outlier detection algorithms. For instance, HDBSCAN can compute outlier scores between 0 and 1 through the density-based GLOSH outlier detection algorithm [27]. GLOSH is computed from a hierarchy a clusters by comparing the density of a point to the densities of the points in the associated current and child clusters. Points with substantially lower density than the cluster density are considered outliers and are given a relatively high outlier score.

Clustering-based anomaly detection methods also suffer from the lack of scalability and the curse of dimensionality, but the test phase is faster as only requires comparison with a few clusters. Thus, more computationally efficient variants are proposed in the literature (see [1,2]) based on heuristic techniques such as in k-means [28], approximate clustering or advanced-indexing techniques to partition the data. A possible way to tackle problems of dimensionality is to first project samples into a smaller dimension space (dimension reduction) before applying clustering techniques on this space. Dimension reduction techniques include PCA [29], t-SNE [30] or autoencoders (see Section 3.3).

### 2.2. Ensemble-Based Methods

Aggarwal [7] provides a broad review of the ensemble-based algorithms for outlier detection and identifies several sub-categories based on a set of principles underlying them. Inside the category of ensemble-based methods, Aggarwal includes classical models for anomaly detection such as local outlier factor (LOF) [13] when they are used with several sets of hyper parameters to combine the resulting scores.

Isolation Forest (IF) [31,32] is another method cited in [7] among the ensemble-based algorithms. This algorithm is specifically designed for anomaly detection with performances challenging sometimes those of the more sophisticated and recent neural network approaches [33]. The method starts building an ensemble of decision trees to classify the data instances. Then, the average of path lengths from the root to the sample location in the trees is computed to determine an anomaly score. The assumption is that anomalies are easier to isolate and so they should have shorter path lengths than normal instances. The method is computationally efficient and can be adapted for application to detect anomalies in streaming data by using a sliding window [34]. More recently, Hariri et al. [35] present a variant of the IF improving the quality of the anomaly scores by correcting the bias induced by the way the branching is done in the classical IF.

### 2.3. Statistical Methods

Statistical anomaly techniques is one of the main categories identified by Chandola et al. [2] in their survey. A similar group of methods is included in the review by Pimentel et al. [1] under the name of *probabilistic novelty detection*. In any case, the authors agree that anomaly detection methods in these categories are based on the estimation of the probability densities of the data and on the assumption that normal data will fall in high probability regions whereas anomalies will fall in low probability ones. The underlying probability distribution is estimated from the training data (assuming it is populated mostly with normal data) and a threshold is set to discriminate anomalous from normal instances.

Both Pimentel [1] and Chandola [2] further classify the methods under the subcategories of parametric and non-parametric techniques and enumerate a multitude of methods which will not

be reviewed here again. Instead we will focus on introducing some of the techniques applicable to time series or widely used by more recent approaches, e.g., Gaussian Mixture Models which are often chosen as prior distributions by the neural network generative models.

### 2.3.1. Gaussian Mixture Models

Gaussian Mixture Models (GMM) are probabilistic models based on the assumption that the instances were generated from a weighted mixture of Gaussian distributions. GMMs can be used for anomaly detection as the distance of a data instance to the estimated mean can be used as an anomaly score. Instances with a score beyond a given threshold are marked as anomalies. GMMs present two main limitations: first they try to fit all the data including the potential outliers in the training set. If the set contains too many outliers, it may hence be useful to remove some of them after a first application of the model. Secondly, in simplest GMM models, the number of Gaussian distributions must be known in advance. Bayesian Gaussian Mixture Model can be used to simplify this process as they eliminate automatically unnecessary clusters.

### 2.3.2. Independent Component Analysis

Independent Component Analysis (ICA) is a statistical technique for data analysis allowing for the identification of latent variables in observed multivariate data. ICA assumes the observed data to be an unknown linear mixture of non-Gaussian and mutually independent latent variables, which are also called independent components, sources or factors.

As an example of application of ICA to anomaly detection, Pimentel et al. [1] refers to Pontoppidan and Larsen [36] who describe a probabilistic framework based on ICA to detect changes in the condition of diesel engines from acoustic emission signals. In aviation, this method have been applied by Jiang et al. [37] to identify air traffic congestion problems (see Section 4).

### 2.3.3. Regression Model-Based

The regression model-based anomaly detection is a subcategory of the parametric techniques identified by Chandola et al. [2] including a number of methods widely applied to time series data. These methods are based on a two-step approach. A regression model is first fitted on the training data. Then the resulting model is used on test sequences to compute the residuals, e.g., the difference between the predicted value and the real value. The anomaly scores are finally determined based on the residuals. Inside this category, we can include anomaly detection techniques based on traditional time series forecasting models such as Vector Auto-Regressive (VAR) [38,39] and Autoregressive Integrated Moving Average (ARIMA) [40,41]. Also, RNN have been used as regression models and will be covered later on in a specific section of the survey.

### 2.4. Domain-Based Methods

This category identified by Pimentel et al. [1] include the methods which define a boundary or domain to separate normal data from anomalies based on the training data. The most widely applied technique in this category and the only one cover in this survey is the Support Vector Machines (SVM) [42] and more precisely the variant known as one-class SVM (OC-SVM) [43].

This method assumes that training data is mostly representative of *normal data* so that the learned boundary properly defines the normal region or class. The boundary is not defined directly on the training data, but in the feature space obtained after applying the kernel trick, i.e., after projecting the data in a space where it is linearly separable. A test instance is then considered as anomalous if falling outside of the defined normal domain.

OC-SVM is part of the Multiple Kernel Anomaly Detection (MKAD) [44] algorithm developed by NASA and considered as one of the first methods proven successful in the detection of anomalies in heterogeneous flight data (see Section 4.1.1).

*2.5. Reconstruction-Based Methods*

Chandola et al. [2] identify a category called spectral-based anomaly detection in which it is assumed that data embedding in a lower dimension helps separate normal instances from anomalous ones. In the taxonomy by Pimentel et al. [1], the reconstruction-based approach encompasses the spectral-based approach (called subspace-based) as well as a neural network-based approach. Reconstruction-based methods assume that anomalies lose information when they are projected to a lower dimension space, hence cannot be effectively reconstructed.

2.5.1. Subspace-Based Methods

In this subcategory, most of the anomaly detection methods use Principal Component Analysis (PCA) [29]. For instance, the surveys [1,2] mention a simple anomaly detection algorithm based on PCA and applied by Dutta et al. [45] to astronomy data. The assumption in this algorithm is that samples with large values for the last principal components (the ones with the lowest variance) are anomalies since this is indicative of a deviation from the correlation structure of data.

Several variants exist to address the different limitations of the basic PCA technique: Kernel PCA [46] introduce specific kernels for non linear projections; Robust PCA [47] aims at making PCA less sensitive to noise by enforcing sparse structures; Functional PCA [48] is a PCA extension [49–51] to the case where data has a functional nature (sample of curves) such as flight trajectories.

PCA models are designed to be trained on a training set: then the fitted linear transformation can be efficiently applied to any volume of further samples. Although the fitted transformations can be applied on datasets of any size, their high computational complexity make them unsuitable to be trained on very large datasets.

The application of subspace-methods as dimensionality reduction techniques are particularly useful when applied to high-dimensional data. Several applications of subspace-methods to aviation exist (see Section 4), including an improved faster method based on Kernel PCA [46] and Functional PCA [52].

2.5.2. Neural Network Methods

Pimentel et al. [1] refer to a variety of NN techniques that can be applied for anomaly detection. We focus here on Autoencoders (AE), one the most widely applied anomaly detection techniques nowadays, which includes variants such as Deep Autoencoder (DAE) or Variational Autoencoder (VAE) (See Section 3).

Autoencoders have the same number of input and output neurons, and one or several hidden layers with a smaller number of neurons acting up as a compression or dimensionality reduction mechanism. The assumption behind reconstruction-based anomaly detection is that anomalies are incompressible and cannot be properly reconstructed from the lower dimensional representation of the latent variables.

Extreme Learning Machines (ELM) [53,54] are feed-forward neural networks much faster to train than SVM or back-propagation neural networks and able to produce good results on many classification and regression problems [55]. They are specially used for scalable anomaly detection in very large datasets: Janakiraman and Nielsen [56] have applied unsupervised ELM models such as autoencoders and embedding models to identify operationally significant events in aviation (see Section 4.1.3).

**3. Recent Advances in Anomaly Detection**

This section reviews some recent techniques applicable to anomaly detection which have been developed in the fields of neural networks or deep learning as well as temporal-logic learning. Table 1 presents an overview of the recent techniques covered in this section.

**Table 1.** Recent Bibliography in Anomaly Detection.

| | | |
|---|---|---|
| **Recurrent Neural Networks** | Section 3.1 | *Stacked LSTM*: [57] (2015)<br>*LSTM and GRU*: [58] (2016)<br>*Hybrid LSTM with OC-SVM or SVDD*: [59] (2017) |
| **Convolutional Neural Networks** | Section 3.2 | *Intrusion detection*: [60] (2017), [61] (2017)<br>*Comparative study with other NN*: [62] (2018) |
| **Advanced Autoencoders** | Section 3.3 | *LSTM-ED*: [63] (2016)<br>*MSCRED*: [64] (2018)<br>*Multi-modal DAE*: [65] (2016)<br>*ConvLSTM-AE*: [66] (2017) |
| **Generative Models** | Section 3.4 | *GAN*: [67], [33], [68] (2018)<br>*Variational Inference*: [69] (2016), [70] (2018) |
| **Temporal-logic Learning Models** | Section 3.5 | *Supervised model*: [71] (2014)<br>*Unsupervised model*: [72] (2014)<br>*Online model*: [73] (2016) |

*3.1. Recent Advances in Recurrent Neural Networks*

A Recurrent Neural Network (RNN) is a special kind of neural network considered as well suited for time series processing. A RNN is called recurrent because it performs the same computation on each element of the input sequence or time series. The output of a RNN depends not only on the current sequence input element, but also on the calculations performed on the previous sequence elements. This memory capability is implemented via a mechanism called hidden state, which is a vector computed at each time step based on current sequence input and the hidden state at the previous time step. The main issue with the standard RNN is its inability to learn long term patterns in sequential data due to the gradient vanishing/exploding problem when applying backpropagation-through-time (BPTT) algorithm during the training phase. For this reason, a standard RNN is rarely used in real world applications which are usually based instead on two improved RNN variants: the Long Short-Term Memory (LSTM) [74] and the Gated Recurrent Unit (GRU) [75].

RNN can be used as a regression model for anomaly detection and as such it can be classified as a method belonging to the regression model-based subcategory identified by Chandola et al. [2] inside the parametric techniques for statistical anomaly detection approaches. Compared to other classical anomaly detection techniques such as the ones based on clustering or OC-SVM, RNN are more convenient to capture temporal and non-linear dependencies in multivariate time series, especially when multiple layers of RNN are stacked together in deep architectures.

Goel et al. [76] perform a comparative study of the performance between two types of LSTM and the traditional Vector Auto-Regressive (VAR) as a regression model for multivariate time series from aviation. Surprisingly, the results of their research show that VAR significantly outperforms LSTM, which according to the authors could be explained by the fact that the LSTM capability of capturing long term dependencies may not be necessary.

Malhotra et al. [57] present a RNN model with several layers of stacked LSTM which is trained with normal data. The trained model is then used as a predictor over a number of multiple time steps and the residuals computed over the training set are modeled as a multivariate Gaussian distribution. The probabilities on the residuals can thus be computed and a threshold to discriminate the anomalies is determined by maximising the $F_\beta$ score over a validation dataset.

More recently, Ergen et al. [59] present a hybrid framework for variable length sequences based on LSTM and the use of either OC-SVM [43] or SVDD [77] as anomaly detectors. The novelty in the approach comes from the fact they jointly optimise the parameters of both LSTM and the anomaly detector by developing specific gradient based training methods. The experiments on several real and simulated datasets show significant performance improvements over the traditional OC-SVM and SVDD methods.

It is also interesting to note that some approaches use RNN as part of an autoencoder (e.g., LSTM encoder-decoder (LSTM-ED) [63,78]) or generative architecture (e.g., GAN-AD [68]). See Sections 3.3 and 3.4 for further details.

*3.2. Recent Advances in Convolutional Neural Networks*

A Convolutional Neural Networks (CNN) is a category of neural networks widely used for image processing. A CNN can be seen as a regularized form of a Fully Connected Network (FCN) since the neurons in one layer are only connected to a small number of neurons in the previous layer rather than to all of them. The regularization approach in CNN is possible because of the hierarchical pattern structure inherent to some kind of data such as images. Thus, a CNN can recognise complex patterns by previously identifying smaller and simpler data patterns via the successive transformations of the input data in the sequence of layers. The vastly reduced number of parameters in the CNN makes them very efficient and their performance in terms of accuracy and generalisation is also relatively better compared to FCN and RNN for tasks such as image classification.

Although RNN have been traditionally considered as the best technique for sequence and time series modelling, recent research [79] seems to suggest that CNN can outperform canonical RNN such as LSTM in the task. CNN are especially well-known for image feature extraction, but they can also be applied to extract complex hidden features in sequential data [80]. In this case, CNN are sometimes combined with some variant of RNN, such as the Convolutional LSTM (ConvLSTM) [81] to better capture spatio-temporal features.

In order to be processed by a CNN, a sliding window is the most used preprocessing technique on time series data. In the case of multivariate time series, some studies [82,83] also suggest computing the pair-wise correlations between the time series to model the system status. The resulting signature matrices can then be fed to a CNN for pattern extraction.

CNN-based anomaly detection methods have been mainly applied to intrusion detection [60,61] by preprocessing data samples with float and integer attributes into an image form convenient for CNN processing. In a more recent study, Kwon et al. [62] assess several CNN architectures for anomaly detection using different network traffic datasets by comparing their performance to other techniques including Variational Autoencoders (VAE), Fully Connected Networks (FCN) [84] and LSTM. Their results indicate that CNN perform better than VAE, but worse than FCN and LSTM.

While the use CNN for anomaly detection is an active area of research, several hybrid architectures integrating CNN exist to perform anomaly detection. This is especially the case when CNN are used as part of more complex AE architectures, like in [64,85,86] which have been applied in the aviation domain and will be further addressed in the next section.

*3.3. Recent Advances in Autoencoders*

Autoencoders (AE) are powerful non linear dimensionality reduction tools commonly used for anomaly detection, with many references in Pimentel's review [1] and more [87]. Autoencoders fall in the unsupervised learning category: they learn to reconstruct, i.e., maximize a similarity measure, samples that go through a lower dimension bottleneck. They *encode*, or *project*, samples into a low dimension representation (the latent space), then *decode*, or *reconstruct*, it back into the original space. The network is trained to minimize the global reconstruction error, e.g., a mean squared error. Once the optimisation converged, anomalies are samples with the higher reconstruction error.

In this section, we focus on recent AE-based anomaly detection approaches including hybrid AE, using RNN or CNN cells in the encoding and decoding parts of the neural networks, as well as Deep Autoencoders (DAE), e.g., Stacked Denoising Autoencoders.

In the case of AE using RNN, it is worth mentioning the research by Malhotra et al. [63]. The authors propose a reconstruction-based approach for anomaly detection in time series based on LSTM Encoder-Decoder (LSTM-ED) models, which has been previously used for machine translation [75]. A LSTM-ED is trained only with normal time series data so that higher reconstruction

errors should be obtained for anomalous sequences. A normal distribution is fitted on the reconstruction errors computed over a subset of the validation data. The estimated mean and covariance is then used to compute an anomaly score for each point in the time series. The threshold to determine whether a point is anomalous or not is computed so that it maximises $F_\beta$ score over a validation dataset.

A ConvLSTM based autoencoder (ConvLSTM-AE) is proposed by [66] to encode appearance and change of appearance (motion) for anomaly detection in videos. More recently, Zhang et al. [64] propose a framework called Multi-Scale Convolutional Recurrent Encoder-Decoder (MSCRED) to perform anomaly detection and diagnosis in multivariate time series data. The architecture combines a convolutional encoder to capture the spatial patterns in the signature matrices, an attention based ConvLSTM to capture the temporal patterns on the previously generated feature maps and finally a convolutional decoder to decode the feature maps obtained in the previous steps in order to get the reconstructed feature matrices.

In [65] a multi-modal Deep Autoencoder (DAE) framework is proposed for anomaly detection and fault disambiguation on multivariate time-series corresponding to flight data generated from multiple sensors. A DAE is a multi-hidden layer autoencoder capable of capturing several levels of data abstraction [88]. In the framework, an overlapping sliding window technique is used over each time series and the resulting sliced time series are concatenated into a large vector and fed into the DAE. The average reconstruction error of a time window over all the sensors is used as an anomaly score. The characterisation of the different fault signatures is based on the analysis of the distribution of the anomaly scores.

### 3.4. Recent Advances in Generative Models

Generative modelling is an area of machine learning which deals with models of distribution defined in some potentially high-dimensional space. Generative models aim at capturing dependencies between dimensions: they are trained to produce realistic data samples looking alike what is in the original data set based on their representation in a lower dimension projected space, commonly referred to as a *latent space*.

Generative Adversarial Networks (GAN) [89] are a well known framework for producing generative models. They consist of two competing networks, a generator and a discriminator. The generator models the data by learning how to transform samples taken from a prior distribution while the discriminator learns to distinguish between real data and samples generated by the generator. GANs have recently been used for anomaly detection [33,67,68,90], also in more advanced variants [91].

Variational Autoencoders (VAE) [92] have also been extensively used for anomaly detection [93]. The neural network representation of VAE is based on traditional autoencoders, although the mathematical foundations have few in common. VAE model high-dimensional distributions by casting learning representations as a Variational Inference [94] problem. VAE aim at learning a mechanism to draw new samples from random variables taking values in the latent space following a fixed prior distribution, classically Gaussian. The optimisation process takes into account the quality of autoencoded samples with respect to their reconstruction probability and the Kullback-Leibler (KL) divergence between the prior distribution and the transformed posterior distribution through the encoding process. An anomaly reconstruct poorly through the generative process, and its encoding fall outside the prior distribution.

VAE based anomaly detection has been generalised to time series by applying RNN with hidden layers as encoder and decoder [95] under the name of Stochastic Recurrent Network (STORN). Similar approach has also been mentioned in [69], or combined with Gaussian Mixtures with Gated Recurrent Unit (GRU) cells [70]. The Gaussian assumption on the prior may be a limitation. Recent approaches attempted to model more complex distributions in the latent space with energy-based models [96] or Gaussian Mixtures Models [97]; and tried to free themselves from the variational framework [98].

More recently, the relevance of VAE over deterministic AE has been discussed [99]: reconstructed examples are often blurry in case of images, the Gaussian assumption on the prior may be too restrictive and the measure of the KL divergence in the optimization problem may lead to over-regularization. The impact of such allegation on anomaly detection has, to our knowledge, not been addressed.

### 3.5. Recent Advances in Temporal Logic-Based Learning

In the previous sections, we have covered the recent advances in anomaly detection in the field of neural networks and deep learning. However, a well known drawback of neural networks is the lack of interpretability of the results. Also, the output of classical methods represented by hyper-planes or surfaces embedded in high-dimensional feature spaces to separate normal from anomalous behaviour is hard to interpret by domain experts. It is true that a decision tree-based method could be applied to identify the rules underlying the anomalies found by an anomaly detection algorithm (not necessarily an isolation forest). However, interpretability in that case would ultimately depend on the number of nodes which increases exponentially with the depth of the tree, and the more the number of nodes, the more difficult it would be to understand the decision rules explaining the anomalies.

Most of the research effort in the last two decades in the field of machine learning and statistics has primarily been focused on designing scalable and accurate black-box models. The interpretability of the results has mostly been neglected because of the general belief it necessarily reduces accuracy [100]. In this section, we introduce a recent anomaly detection approach that can learn from data temporal logic properties of a system in the form of a more human-readable formalism based on temporal logic expressions. This approach can be better accepted by domain experts who intrinsically dislike black-box models and ultimately reject them because of their lack of transparency.

Thus, Jones et al. [72] and Kong et al. [71,73] present an approach capable of inferring signal temporal logic (STL) [101,102] formulae from data resembling natural language. STL is a specification language used in the field of formal methods to specify system properties including time bounds and bounds on physical system parameters, which can be used to describe the normal system behaviour. For instance, we can express invariant properties such as "If x is greater than $x_r$, then within $T_1$ seconds, it will drop below $x_r$ and remain below $x_r$ for at least $T_2$ seconds" [71].

The original supervised method [71] and unsupervised method [72] have been recently extended to allow for on-line anomaly detection [73]. This new algorithm has reduced the computational cost compared to the supervised version [72] and can be now applicable to high dimensional systems producing large amounts of data. A further advantage is that the output is expressed in STL, which can be directly processed by a computer system for automatic monitoring of anomalous behaviour.

The approach has been applied to several domains including naval surveillance and train braking system [73]. Concerning the aviation domain, Deshmukh et al. [103] has recently used the approach to detect anomalies in the terminal airspace operations (more details in Section 4.1).

## 4. Applications

This survey reviews the use of some of the previously introduced anomaly detection methods in two important areas of the aviation: air traffic operations and predictive maintenance. Because of the significant number of the techniques covered in the first application area, air traffic operations, we have created a classification of the methods based on the category of the anomaly detection approach (see Table 2).

**Table 2.** Application of anomaly detection models to aviation use cases.

| | |
|---|---|
| **Section 4.1.1 Domain-based** | *Abnormal approaches with MKAD*: [104] (2011)<br>*GA approach and landing anomalies with OC-SVM*: [105] (2017) |
| **Section 4.1.2 Distance-based** | *Anomalous pilot switching with SequenceMiner*: [18] (2008)<br>*Anomalous take-off and approach operations*: [19] (2011), [20] (2015)<br>*Anomalous safety events with LoOP*: [16] (2019)<br>*Anomalous taxi paths with hierarchical clustering*: [22] (2019)<br>*Anomalous radiotelephony readbacks with kNN*: [106] (2018) |
| **Section 4.1.3 Reconstruction-based** | *Atypical aviation safety data with KPCA*:  [107] (2017)<br>*Atypical approaches and landings with FPCA*:  [52] (2018)<br>*Anomalous trajectories in TMA and en-route*: [108] (2018), [109] (2019)<br>*Anomalous transitions between sector configurations*: [110] (2018)<br>*Anomalous ADS-B messages with ConvLSTM-AE*: [86] (2019) |
| **Section 4.1.4 Statistical-based** | *Anomalous flights with VARX*: [38] (2016),<br>*Anomalous flight switches with VAR*: [39] (2016)<br>*Abnormal flight data with GMM*: [21] (2016)<br>*Anomalous air traffic congestion with ICA*: [37] (2019) |
| **Section 4.1.5 Temporal-logic based** | *Anomalous trajectories in terminal airspace with TempAD*: [103,111] (2019) |

## 4.1. Anomaly Detection for Air Traffic Operations

One application area in aviation where anomaly detection techniques have particularly been applied to is in the identification of significant operational events in flight data. In this context, significant events mean patterns or behaviours that can be worth detecting in flight data because of their potential impact on the performance (usually safety) of flight operations. For instance, the identification of events such as runway excursions, go-around operations, trajectory deviation due to conflict resolution actions. Other significant events occur in the broader context of Air Traffic Management (ATM) operations. These are also covered in this section and include anomalous ATC-pilot communications and anomalies in the sequences of airspace sector configurations. Table 2 presents an overview of the different applications classified by the category of the anomaly detection approach applied.

In the US, NASA established in 2007 a program to store Flight Operations Quality Assurance (FOQA) data from most of the major airlines to monitor and address operational risk issues. FOQA data is now managed as part of a FAA program called Aviation Safety Information Analysis and Sharing (ASIAS), which integrates a variety of information sources from different parts of the NAS. The FOQA database currently contains millions of flights and each entry represents hundreds of parameters from the avionics and other on-board systems. Likewise, in Europe, Flight Data Monitoring (FDM) programs promoted by EASA requires airlines to gather, monitor and analyse data to improve the performance and safety of flight operations.

The objective of FOQA and FDM programs is to switch from a purely reactive mode based on reports or interviews to a more proactive mode where data analytics can be used to assess trends, risks and undesired events in order to help implement mitigation measures. The applications reviewed here support this proactive approach by automatically detecting statistically anomalous events in vast amounts of historical on-board generated data.

However, the process is not fully automatic as the flagged events need further consideration from operational experts to determine whether the identified anomalies are only low occurrence events or true significant events with potential safety or performance implications.

For decades, the only approach to automatically detect anomalies from generated data has been based on exceedance detection algorithms, which check flight data against predetermined thresholds set by subject matter experts. When one or a combination of thresholds are exceeded, the corresponding flight is flagged as anomalous. Even though this approach has been improved and is nowadays largely trusted by the industry, it still presents significant shortcomings such as the difficulty to properly set

the thresholds to avoid false-positives and false-negatives as well as the impossibility to anticipate all possible events.

The availability of extensive amounts of generated flight data along with the significant advances in the machine learning community offer new opportunities for approaches capable of a better detection of unknown (not pre-programmed) events, which should improve the current rate of false-negatives in exceedance-based approaches and be able to cope with a large volume of high-dimensional data. In the following subsections, we present the application of some of the previously reviewed data-driven anomaly detection methods (see Sections 2 and 3) to support the identification of significant flight operational events.

### 4.1.1. Domain-Based Approaches

If SequenceMiner [18] is one of the few anomaly detection methods specifically designed for the processing of discrete sequences, MKAD developed by Das et al. [44] is one of the first methods designed to effectively detect operationally significant anomalies with heterogeneous sequences of both discrete and continuous variables. Based on kernel functions and OC-SVM, MKAD can identify operational situations in FOQA data such as go-around operations, unusually high airspeed flights, flights impacted by gusty winds and abnormal approaches. More recently, Das et al. [104] applied MKAD to detect anomalies in the approach phase but this time with a much larger set of flights of the same fleet and aircraft type. In the paper, the authors report exclusively on two anomalous situations correctly identified by MKAD corresponding to two significant operational events: high energy approaches and turbulent approaches.

With the aim of improving the safety of General Aviation operations, Puranik et al. [105] propose a framework to identify anomalies based on a OC-SVM model. After a classical preprocessing phase to clean the raw multivariate time series data, a set of feature vectors corresponding to the energy metrics detailed in [112] are computed, such as the Specific Total Energy (STE) or the Specific Potential Energy (SPE). The DBSCAN algorithm is first applied to the feature vector in order to determine the number of clusters. Based on the identified clusters, the OC-SVM algorithm is used to compute the anomaly scores of each flight. The methodology is evaluated with both simulated data with anomalies and real data from a Cessna 172S during the approach and landing phase. The results show a good performance in terms of anomalous flight identification even when only a limited number of parameters are recorded.

### 4.1.2. Distance-Based Approaches

One of the first attempts in the field was the research by Budalakoti et al. [18] who address the problem of anomaly detection in a set of sequences of switches used by the pilot and co-pilot to maneuver an aircraft. Their method (SequenceMiner), based on a clustering approach, is able to detect anomalous switching behaviour linked to the loss of autopilot mode awareness by the flight crew.

Li et al. [19] apply a cluster-based anomaly detection (ClusterAD) method based on DBSCAN to detect anomalies in a FOQA dataset of an airline for 365 B777 take-off and approach operations. The anomalous operational situations correctly identified by ClusterAD include high/low energy approaches, unusual pitch excursions, abnormal flap settings and high wind conditions. One of the advantages of ClusterAD compared to MKAD is that it can automatically identify multiple types of flight operation patterns (different nominal operations) corresponding to the identified clusters.

Following up this research, Li et al. [20] present a method based on DBSCAN called ClusterAD—Flight, which is able to detect abnormal flights during take-off or approach as whole. In this work, more extensive tests are conducted with an additional dataset of 25,519 A320 flights. Results show that both ClusterAD—Flight and MKAD are able to identify more operationally significant anomalies than exceedance-based methods. ClusterAD—Flight performs better with continuous parameters, whereas MKAD is more sensitive toward discrete parameters. The latest research by

Li et al. [21] is on an improved ClusterAD approach called ClusterAD—DataSample. However, as this method is based on a GMM, we cover it as part of the statistical approaches in Section 4.1.4.

Compared to MKAD and ClusterAD which are able to process hundreds to tens-of-thousands of flights, Oehling et al. [16] propose an approach able to scale to very large datasets as the ones used in the production environments of big airlines. The approach, based on the Local Outlier Probability (LoOP) method, is applied to an airline dataset of 1,2 million flights in order to detect anomalies related to safety events. The top outliers identified by their approach are reviewed by the airline pilots in order to assess their safety-relevance. The results of the research show that their method is able to reduce the number of undetected safety-relevant events compared to the current exceedance based approaches implemented in FDM systems.

Churchill et al. [22] present a hierarchical clustering method to group in space and time aircraft trajectories in the airport surface. The goal is the identification of statistically anomalous taxi paths, which may be unplanned and unexpected by the controllers and thus could represent a safety risk.

In [106,113], semantic checking models based on LSTM and kNN are introduced to identify read-back errors in ATC radio-telephony communications. Civil aviation radio-telephony recordings are converted to textual format, and similarity functions are defined to verify whether the semantics is the same between controller instructions and pilot read-backs.

### 4.1.3. Reconstruction-Based Approaches

Zhang et al. [107] point out two known issues when the classical Kernel PCA algorithm [46] is applied to a large dataset for anomaly detection: it is computationally expensive ($O(n^3)$) where $n$ is the size of the dataset) as well as hard to adapt as parameters such as the number of principal components and the confidence for the confidence limit needs to be set before anomaly detection. Thus, the authors develop an optimized GPU implementation where the previous parameters are computed automatically. The improved algorithm is applied to synthetic datasets [44] and compared to the OC-SVM [43] technique. The results show significant speed increases and a detection efficacy close to the OC-SVM one.

Jarry et al. [52] propose a method based on FPCA to identify atypical approaches and landings both in post-operational analysis and on-line. The method was tested with track radar data (20,756 records) of landing operations at Paris Charles-De-Gaulle (CDG) airport. The goal is to improve the detection rate of Non Compliant Approaches (NCA), i.e., an approach in which the intercepting conditions of the intermediate and final legs are not compliant with the operational prescriptions. NCA is a precursor of Non Stabilised Approaches (NSA) which may lead to fatal events like Control Flight Into Terrain (CFIT). The authors propose to extend current tools capabilities based on geometric criteria by taking into account additional features such the specific total energy of the aircraft. The method uses a sliding window over the trajectories in order to apply FPCA first and then HDBSCAN with GLOSH [27]. From the set of computed outlier scores, it is determined whether a trajectory is anomalous. The results show the method can effectively identify atypical flights although the results can be very sensitive to the size of the sliding window.

Janakiraman and Nielsen [56] propose an unsupervised anomaly detection approach based on ELM. This approach developed by NASA is an alternative to MKAD for the identification of safety risks in very large aviation datasets. The performance of the three ELM variants are evaluated and compared to MKAD on a dataset of over 40,000 flights corresponding to landing operations at Denver airport. While the results of the ELM-based approach are comparable to MKAD in terms of detection accuracy, the training of ELM models are faster by two orders of magnitude.

Olive et al. [108] present a method based on autoencoders to analyse flight trajectories, detect unusual flight behaviours and infer ATC actions from past Mode S data. The method is evaluated with three different city-pairs and one year of traffic within a bounding box defined just before the entry to the Terminal Manoeuvring Area. The identified anomalous situations are analysed based on the distribution of reconstruction errors (anomaly scores). It is shown that the highest anomaly scores

correspond to weather impact or traffic regulations whereas the lowest ones to relatively more usual ATC deconfliction or sequencing actions.

Following up on the previous research on air traffic anomaly detection [108] and on identification of traffic flows [114] in en-route ATC sectors, Olive and Basora [109] propose a method to detect anomalous flight trajectories in the flows of a en-route ATC sector. A clustering approach is used first to automatically identify from ADS-B traffic a set of clusters corresponding to sector flows. Then, an autoencoder is applied to each cluster in order to detect anomalous trajectories. The analysis of the distribution of reconstruction errors confirms the conclusions reached in [108].

In [110], autoencoders are used to detect anomalous transitions between sector configurations in Area Control Centres (ACC). The model is trained with transitions performed in the past and then applied to transitions never realized. Transitions with highest autoencoder reconstruction error are considered as anomalies, unlikely to be realized.

Based on the ConvLSTM method by Shi et al. [81], Akerman et al. [86] present a convolutional LSTM based autoencoder (ConvLSTM-AE) framework to detect anomalous ADS-B messages. In this framework, aircraft flying in the same airspace are represented as images and the ConvLSTM-AE model is used to detect anomalies in the sequences of images leading to anomalous ADS-B location reports.

### 4.1.4. Statistical-Based Approaches

Melnyk et al. [38] propose an unsupervised model-based framework adapted to online anomaly detection where each flight is represented as a Vector AutoRegressive eXogenous model (VARX) model [115]. The key step in the approach is to compute a distance matrix between flights defined in terms of residuals of modeling one flight's data using another flight's VARX model. Once the distance matrix is built, a LOF method [13] is applied to identify the anomalous flights. The evaluation results on a large FOQA dataset (over a million flights) show a good performance into detecting already known safety events as well as previously undetected ones compared to state-of-the-art algorithms like MKAD.

In another framework also based on VAR modelling and adapted to online anomaly, Melnyk et al. [39] represent each flight with a semi-Markov switching vector autoregressive (SMS-VAR) model. With this approach, each phase of a flight determined by the set of pilot switches is represented by a different VAR process [115] and a semi-Markov model (SMM) [116] is used for the dynamics of flight switches. Anomaly detection is based on the dissimilarities between the one-step ahead model's predictions and observed data. The framework is extensively evaluated on both a synthetic and an airline FOQA dataset and the achieved performance is similar or slightly better than the MKAD one.

Nanduri and Sherry [58] present a regression-based approach applied to simulated FOQA-like data [117] corresponding to 500 approaches into San Francisco airport. Several different kinds of RNN architectures (GRU and LSTM) are tested and compared with MKAD. In all cases, the RNN-based models are able to detect more anomalies than MKAD. The authors explain the superiority of RNN-based approaches by the fact that they do not have the limitations of MKAD: the need for dimensionality reduction which results in loss of information and poor sensitivity to short duration anomalies, and its inability to detect anomalies in latent features. Unfortunately, the authors give few details on how the anomaly threshold based on the residuals (prediction errors) was chosen.

ClusterAD—DataSample by Li et al. [21] is a method based on a GMM which is capable of instantaneously detecting abnormal data samples during a flight rather than abnormal flights as a whole during a specific flight phase. The method is tested with a real dataset of 10,528 A320 flights and compared with exceedance-based methods. Then, it is compared with MKAD and ClusterAD—Flight with a second dataset of 25,519 A320 flights (already used in [20]). The results indicate ClusterAD—DataSample performs better in detecting known unsafe events (detected with exceedance-based methods), but the authors point out the need for further evaluation of the performance with detecting unknown issues.

Jiang et al. [37] propose a method based on independent component analysis (ICA) for online monitoring of air traffic congestion. Based on the complex networks topology, a model is trained with a dataset of smooth situations. Any new situation is then compared to the reference 'normal' representation by analyzing the change of statistics. As the confidence limits cannot be determined directly from a particular approximate distribution, a kernel density estimation (KDE) is used to set the control limits.

### 4.1.5. Temporal-Logic Learning Based Approaches

Deshmukh et al. [103] propose a temporal logic based anomaly detection algorithm (TempAD) applicable to trajectories in the terminal airspace. The algorithm, based on a temporal-logic learning approach [71–73], can learn human-readable mathematical expressions from data which facilitates the feedback and interaction with operational experts. The method uses DBSCAN as a preprocessing step to identify the clusters with similar trajectories on which the detection of anomalies with TempAD becomes more effective. TempAD is able to generate for each cluster STL predicates defining the bounds of normal flights as a function of time, distance to touchdown or aircraft state vectors (including latitude, longitude, altitude, ground speed). The representative features to find anomalies include some of the energy features used in [105]. The method is evaluated on real surveillance data from the terminal airspace at New York La Guardia airport, covering several thousands of arrival flights. The algorithm is able to effectively identify anomalous situations such go-around operations as well as arrivals with excessive total energy, above or below the recommended glideslope.

Following up this research, Deshmukh et al. [111] develop a supervised precursor detection algorithm called reactive TempAD by correlating surveillance data to specific anomalies identified by the TempAD algorithm [103]. Thus, the prediction of an anomaly is performed by identifying events that precede the occurrence of an anomaly, which are called precursors.

### *4.2. Anomaly Detection for Predictive Maintenance Operations in Aviation*

Flight data recorders generate large volumes of heterogeneous time-series data from arrays of sensors. This massive amount of sensor-generated data can be exploited to perform fault diagnosis and estimate the remaining useful life (RUL). The long-term objective is to reduce and ultimately avoid unscheduled maintenance by optimising the scheduling of maintenance operations based on the RUL prediction, i.e., condition based maintenance (CBM). The ability to predict the RUL of a system component after the occurrence of a fault corresponds to the widely accepted definition of prognostics. The field of prognostics and health management (PHM) has drawn significant interest from industrial and academic research in the last few years as system availability and reliability becomes a serious concern, especially in safety-critical systems such the ones found in aviation.

In the emerging field of data-driven prognosis, predictive models are learned from flight and maintenance data. These models can then be integrated into PHM systems for health monitoring and incipient system failure prediction. There exists a number of data-driven methods for prognosis, but it is usually difficult to compare them based on a common reference baseline due to the use of sensitive commercial data. Fortunately, the following open datasets related to aviation are widely acknowledged as reference for comparison:

- NASA DASHlink open database originally designed and collected by Balaban [118] and available at https://c3.nasa.gov/dashlink/projects/85/;
- a turbofan engine degradation simulation dataset based on thermo-dynamical simulation models, introduced in [119];
- other datasets, also shared on the Prognostic Data Repository of NASA refer to bearing systems or milling machines. These do not necessarily refer to aviation problems but are still worth mentioning as they are commonly used as reference.

Although prognostics and RUL estimation is a core function of PHM, it falls out of the strict anomaly detection scope and hence it is not covered in this survey (two recent reviews by Elattar (2016) [120] or Lei (2018) [121] already address this topic). Instead, we focus our review on the application of data-driven techniques aimed at detecting anomalous behaviour in aviation systems with the goal of identifying faults after their occurrence or anticipating potential failures as part of the condition monitoring process in PHM [122].

Effective anomaly detection techniques to predict incipient failures from historical data is important to estimate time-to-failure and help schedule maintenance activities. Some of the reviewed techniques can also support fault diagnostics, which is a PHM process encompassing fault detection, isolation (i.e., which component has failed), failure mode identification (i.e., what is the cause of failure or fault) and quantification of the failure severity. Fault detection is typically based on the quantification of the inconsistencies between the actual and the expected behavior of the system in nominal conditions [122].

For instance, Rabatel et al. [123] present an anomaly detection framework for preventive maintenance based on anomalous pattern detection in data. The data is based on closed railway data; the approach, from pattern extraction to anomaly detection methods to apply on sequences, could be extended to aircraft data which are subject to common characteristics.

More recently, Nicchiotti et al. [124] (2018) leverage closed commercial aircraft maintenance operational data and apply SVM based methods and PCA as a tool to reduce dimensionality in order to predict such unscheduled maintenance operations.

Deep autoencoders [65] and convolutional denoising autoencoders [85] (2019) have been used for fault detection and anomaly detection, both on the NASA open database and on a dataset of Customer Notification Reports sent over ACARS to airlines to help them detect engine faults.

Recurrent Neural Networks autoencoders have also been used on time series [125] in order to find a proper embedding or representation of time series that is in turn used for predicting a RUL estimation on the turbofan dataset. More recently, Zhao et al. compare in [126] different approaches of feature selection mechanism based on dimensionality reduction. Autoencoders, Riemann Boltzmann Machines (RBM) and Deep Belief Networks (DBN), CNN and RNN based methods are compared on a traditional milling machine health monitoring application with similar results, the older RBM/DBN-based techniques being slightly behind.

## 5. Conclusions

In this survey, we have reviewed the state-of-the-art in data-driven anomaly detection and its application to the aviation domain. Thus, we have introduced a large number of classical and more recent approaches and described how some of them have been applied to areas such as air traffic operations and predictive maintenance. Machine learning models can work with offline or online data to detect significant events for further analysis by aviation experts as part of decision support or condition monitoring tools. The ultimate goal of the presented applications is to help improve the performance of ATM and maintenance operations, in particular safety.

In general, as stated by Janakiraman and Nielsen [56], the data-driven detection of anomalies in aviation data is particularly challenging because of its large volume, high-dimensionality, heterogeneity (mixed categorical and continuous attributes), multi-modality (multiple modes of nominal and non-nominal operations with different types of aircraft, airports and airspaces) and temporality (long time-series). The challenge is expected to be even bigger in the future because of the forecast world-wide growth of air traffic and the ever higher number of sensor-equipped aviation systems and operational complexity.

Classical nearest-neighbour and clustering-based approaches do not scale well with such massive amounts of high-dimensional data. In the case of high-dimensional data, the use of a dimensionality reduction technique as a preprocessing step (e.g., to clustering) or the application of a reconstruction-based method is often a better solution. On the other hand, distance-based methods are

computationally expensive when applied to large volumes of data, even during the test phase, which makes them unsuitable for real-time applications. However, in the case of probabilistic, domain-based and reconstruction-based methods, even though the training phase can be time-consuming, the test phase is very efficient. This is not an issue for applications where models can be trained offline, but some real-time safety monitoring applications may require some kind of incremental or very fast online training.

In aviation, among the traditional approaches, the domain-based MKAD [44] developed by NASA is still one of the state-of-the-art methods for the detection of operationally significant events in flight data. However, its computational complexity is quadratic with respect to the number of training examples, which makes it unsuitable for very large datasets and certain applications. ClusterAD methods are also among the most widely applied, in spite of having the same performance issues than MKAD. For faster learning with large datasets, ELM [56] or LoOP [16] based anomaly detection seem to be two good alternatives to MKAD.

The recent advances in anomaly detection we have covered in this review are mainly based on techniques developed in the field of neural networks and deep learning. In principle, deep-learning approaches should be better adapted than traditional machine learning methods [11] when it comes to find anomalies in large-scale high-complex data as the one generally available in aviation. We have also reviewed the advances in temporal-logic based learning as an alternative approach that should help the user more naturally understand and trust the results expressed in terms of logical formulae.

The application area related to the identification of significant events in air traffic operations is particularly rich in terms of the number and variety of the anomaly methods applied. While traditional techniques are widely used, there exists also some attempts to apply recent advances in temporal-logic learning [103,111], RNN [58] and advanced autoencoders (e.g., ConvLSTM-AE [86]). The vast majority of the research in this application area concern the detection of anomalies relevant to safety, although we provided also a few examples of anomalies related to potential cyberattacks or air traffic congestion. Another observation is that most of the introduced applications work in an offline configuration with post-operational data for analysis purposes rather than with online data to support real-time monitoring tasks.

As for the other application area concerning predictive maintenance in aviation, we have reviewed a few anomaly detection methods aimed at identifying incipient failures in aviation system components from flight and maintenance data. These data-driven methods play an increasingly important role in PHM which is necessary to achieve true condition-based maintenance. In spite of that, the number of reviewed research is relatively limited compared to the air traffic operations application area. This is because a lot of the literature on data-driven methods for PHM is more focus on RUL prediction which is out of the scope of this review. Also, some of the work on anomaly detection for predictive maintenance is not specific to aviation. Nevertheless, we have covered the application of both classical approaches such as SVM [124] and more recent approaches based on deep learning such as deep autoencoders [65].

Finally, the operational usability of anomaly detection methods as part of a decision support tool is an aspect marginally addressed and which would probably deserve further attention and research. A first consideration about the usability of anomaly detection methods is how to provide the user with a proper uncertainty measure associated to the model output (e.g., confidence intervals) as a better way to deal with false alarms. A second consideration is that for an expert to trust and understand the prediction of an anomaly detection model, the model and its outputs should be explainable in some degree. Although this issue is more generally addressed in an emerging research field called explainable artificial intelligence [127], its main focus has been on supervised machine learning approaches, which is not the main approach in anomaly detection.

**Author Contributions:** Conceptualization, L.B. and X.O.; methodology, L.B. and X.O.; software, n/a; validation, L.B., X.O. and T.D.; formal analysis, L.B., X.O. and T.D.; investigation, L.B., X.O. and T.D.; resources, L.B., X.O. and T.D.; data curation, n/a; writing—original draft preparation, L.B., X.O. and T.D.; writing—review and editing, L.B., X.O. and T.D.; visualization, n/a; supervision, L.B.; project administration, X.O.; funding acquisition, X.O.

**Funding:** This research has received funding from the European Union's Horizon 2020 research and innovation programme under grant agreement No 769288.

**Conflicts of Interest:** The authors declare no conflict of interest.

## Abbreviations

The following abbreviations are used in this manuscript:

Abbreviations related to machine learning methods

| | |
|---|---|
| AE | Autoencoder |
| ARIMA | Auto Regressive Integrated Moving Averages |
| CNN | Convolutional Neural Network |
| DAE | Deep Autoencoder |
| DBN | Deep Belief Network |
| DBSCAN | Density-Based Spatial Clustering of Applications with Noise |
| ELM | Extreme Learning Machines |
| GAN | Generative Adversarial Network |
| GLOSH | Global-Local Outlier Score from Hierarchies |
| GMM | Gaussian Mixture Model |
| GRU | Gated Recurrent Unit |
| ICA | Independent Component Analysis |
| IF | Isolation Forest |
| KDE | Kernel Density Estimation |
| kNN | K-Nearest Neighbours |
| IMF | Inductive Monitoring System |
| LOF | Local Outlier Factor |
| LoOP | Local Outlier Probability |
| LSTM | Long Short-Term Memory |
| MKAD | Multiple Kernel Anomaly Detection |
| NN | Neural Network |
| OC-SVM | One-Class Support Vector Machine |
| OPTICS | Ordering Points To Identify the Clustering Structure |
| PCA | Principal Component Analysis |
| RBM | Riemann Boltzmann Machine |
| RNN | Recurrent Neural Network |
| STORN | Stochastic Recurrent Network |
| SVM | Support Vector Machine |
| VAE | Variational Autoencoder |
| VAR | Vector Auto-Regressive |

Abbreviations related to aviation

ACARS    Aircraft Communication Addressing and Reporting System
ADS-B    Automatic Dependent Surveillance–Broadcast
ASIAS    Aviation Safety Information Analysis and Sharing
ATC      Air Traffic Control
ATM      Air Traffic Management
CBM      Condition Based Maintenance
FDM      Flight Data Monitoring
FOQA     Flight Operations Quality Assurance
NAS      National Airspace System
PHM      Prognostics and Health Management
RUL      Remained Useful Life

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
