# Peer review of "Recent Advances in Anomaly Detection Methods Applied to Aviation"

_aerospace, doi:10.3390/aerospace6110117_

Round 1
Reviewer 1 Report
The authors provided an extensive and comprehensive overview of anomaly detection methods in aviation. The papers reviews all essential state-of-the-art methods and discusses relevant applications of these methods in the areas of ATM and predictive maintenance. The paper is well written and structured. The paper is relevant and would be interesting to the readers of the journal.
A few suggestions for improvement:
1. Could some criteria for selection/comparison of different methods be introduced? (e.g., in terms of interpretability, computational complexity, generalisation capabilities etc)
2. The interpretability of the learned models is considered in Section 3.5. Could the authors comment on the interpretability of the decision tree-based methods?
3. The applicability of the techniques for particular types of data (i.e., coupling between different types of data and techniques) could be discussed.
Minor comments:
P9, Line 351: please explain what RNN are
P9, Line P384: please introduce CNN
P9, line 335: ‘autoencoders’
Author Response
The authors would like to thank the reviewer for the suggestions. Indeed the questions raised are really interesting to discuss and we are looking forward to reading future research works about these questions.
More specifically,
We mentioned the issues of interpretability in Section 3.5 about temporal logic, of computational complexity with several methods including PCA or MKAD, many of them being quadratic as mentioned in the manuscript. About generalisation, we are not aware of existing criteria to evaluate generalisation for unsupervised learning, in the absence of labels. We included comments about generalisation in the additions coming from the minor points you raised below. About interpretability of resulting models, we added a paragraph in Section 3.5 About applicability of techniques to particular types of data, we adopted an orthogonal point of view, assuming that most data we consider are time series, and chose to present different techniques depending on the application in Section 4. We also discussed the issue of dimensionality, volume and multi-modality along the manuscript and in the conclusion. (see for example paragraph starting line 687, or starting line 787)About minor points, you will find the amendments in blue in the pdf file. You will find them in pages 9, 10 and 11.
Yours sincerely
Reviewer 2 Report
This paper is, overall, a well-written survey on anomaly detection applied to aviation. It covers a large range of methods, operational aviation data, and types of anomalies.
There are a few points I want to raise:
On page 2 is written, "In a semi-supervised approach it is assumed that the training set contains only normal data. On the other hand, unsupervised learning techniques assume only that there is a small enough fraction of anomalies in the data so as to avoid a high rate of false alarms." This is inaccurate. In semi-supervised learning, we assume that there are examples of more than one type, although many examples are unlabeled and are assumed to have the same labels as the labeled examples that are nearby in terms of distribution. In unsupervised learning for anomaly detection, we assume that a small fraction of data that are far away from most data points or are in low-density regions are anomalous. On page 13, the NASA program of FOQA data from multiple airlines that is now operated by the FAA is called Aviation Safety Information Analysis and Sharing (ASIAS). I don't have a reference handy. Please edit the paper for typographical errors and fix the references (e.g., 18, 22, 44).Author Response
The authors would like to thank the reviewer for noticing these inexact points in the manuscripts.
More specifically:
page 2/3: we included your comments about semi-supervised learning in the revised version of the PDF (see before/after version in blue) page 13 (now page 14): we included your precision about FOQA in the manuscript (also in blue). Thank you for the update. We fixed many references, specifically 23, 25, 26, 30, 33, 53, 63, 64, 65, 67, 68, 69, 70, 76, 79, 92, 99, 108, 111 and 114.Yours sincerely
Round 2
Reviewer 1 Report
The authors have sufficiently addressed my comments.